# High Accuracy Indicators of Androgen Suppression Therapy Failure for Prostate Cancer—A Modeling Study

**DOI:** 10.3390/cancers14164033

**Published:** 2022-08-20

**Authors:** William Meade, Allison Weber, Tin Phan, Emily Hampston, Laura Figueroa Resa, John Nagy, Yang Kuang

**Affiliations:** 1School of Mathematical and Statistical Sciences, Arizona State University, Tempe, AZ 85281, USA; 2School of Computing and Augmented Intelligence, Arizona State University, Tempe, AZ 85281, USA; 3College of Computing, Georgia Institute of Technology, Atlanta, GA 30332, USA; 4Theoretical Biology and Biophysics Group, Los Alamos National Laboratory, Los Alamos, NM 87545, USA; 5Department of Mathematics, State University of New York, Buffalo, NY 14260, USA; 6Department of Life Sciences, Scottsdale Community College, Scottsdale, AZ 85256, USA

**Keywords:** mechanistic model of prostate cancer, predictive modeling, evolutionary cell quota framework, adaptive cancer management, dynamic indicator of treatment failure

## Abstract

**Simple Summary:**

Hormonal therapy for prostate cancer is often applied past the point of resistance, hence losing any future clinical value to the evolution of resistant strains. If the undesirable outcome of the treatment is forewarned, then clinicians can have an opportunity to adjust the treatment, which can result in better management of the cancer. Using a mechanistic mathematical model, we introduce two methods to enhance the accuracy of classical biomarkers for hormonal therapy failure. Our results show the value in measuring both prostate-specific antigen and androgen during hormonal treatment, which can potentially allow for better management of prostate cancer.

**Abstract:**

Prostate cancer is a serious public health concern in the United States. The primary obstacle to effective long-term management for prostate cancer patients is the eventual development of treatment resistance. Due to the uniquely chaotic nature of the neoplastic genome, it is difficult to determine the evolution of tumor composition over the course of treatment. Hence, a drug is often applied continuously past the point of effectiveness, thereby losing any potential treatment combination with that drug permanently to resistance. If a clinician is aware of the timing of resistance to a particular drug, then they may have a crucial opportunity to adjust the treatment to retain the drug’s usefulness in a potential treatment combination or strategy. In this study, we investigate new methods of predicting treatment failure due to treatment resistance using a novel mechanistic model built on an evolutionary interpretation of Droop cell quota theory. We analyze our proposed methods using patient PSA and androgen data from a clinical trial of intermittent treatment with androgen deprivation therapy. Our results produce two indicators of treatment failure. The first indicator, proposed from the evolutionary nature of the cancer population, is calculated using our mathematical model with a predictive accuracy of 87.3% (sensitivity: 96.1%, specificity: 65%). The second indicator, conjectured from the implication of the first indicator, is calculated directly from serum androgen and PSA data with a predictive accuracy of 88.7% (sensitivity: 90.2%, specificity: 85%). Our results demonstrate the potential and feasibility of using an evolutionary tumor dynamics model in combination with the appropriate data to aid in the adaptive management of prostate cancer.

## 1. Introduction

Prostate cancer is the most prevalent cancer among men in the US, and therefore is a significant public health concern [1]. Treatment for prostate cancer has advanced considerably over the past several decades [2], but treatment resistance remains a significant threat to every existing therapy. The high degree of heterogeneity in prostate cancer means that even treatments with a promising initial response can fail when the neoplasm inevitably evolves to become treatment-resistant [3,4]. Furthermore, applying any therapy to the point of failure only serves to further fortify the existing resistance. If it were possible to identify an incipient resistance then a clinician would have an opportunity to change treatment strategy to potentially reach a more favorable outcome of the overall regimen [5,6,7].

By nature, a defining characteristic of cancer is a highly unstable genome. As a result, the constituent cells of any neoplasm are endlessly differentiating [8,9,10]. Therefore, tumors tend not to be homogeneous collections of genetically identical cancerous cells [10]. New genomic variations arise continually and rapidly, and some inevitably confer traits that allow them to evade a previously effective therapy [10,11]. When applied to resistant cancer cells, treatment selects for the dominance of the resistant trait. Once the susceptible cells are eliminated, the tumor becomes permanently and irrevocably resistant to that treatment. This clonal model of resistance explains why it is undesirable to continue any therapy to the point of failure [10,12].

There is evidence that treatment resistance comes at the expense of fitness, which means resistant phenotypes are unlikely to become dominant in a treatment-free environment [13]. Therefore, it has been suggested that one could exploit intercellular competition by adjusting the timing and intensity of treatment to effectively reduce the development of treatment-resistant phenotypes and thereby manage the progression of the tumor. This is the central idea of adaptive therapy, deeply rooted in ecological theory, but difficult to execute in practice [6,7,13].

Pretreatment, prostate cancers are androgen dependent. Androgens diffuse into prostate cells and bind to intracellular androgen receptors, which then activate proliferation and survival pathways in both healthy and cancerous prostate cells. For this reason, androgen deprivation therapy (ADT) is the standard of treatment for advanced or metastatic prostate cancer [14]. The therapy uses an agonist and/or antagonist of luteinizing hormone-releasing factor combined with antiandrogen drugs to eliminate primary androgen production in the testes. ADT is initially effective at stopping and reversing the growth of the tumor, but treatment resistance inevitably arises [1,15,16].

The standard application of ADT involves continuously applying the treatment at maximum dosage to eradicate the tumor quickly (continuous androgen suppression, or CAS). However, due to the adverse side effects of ADT and the imminent development of treatment resistance, intermittent androgen suppression therapy (IAS) was theorized to be the better alternative. IAS is a rigid form of adaptive therapy, where ADT is applied at maximum dosage in intervals, cycling on and off treatment in periods of fixed duration or timed based on the growth of the tumor [17]. IAS has several advantages over CAS. Most notably, the off-treatment periods in IAS give patients a break from the adverse side effects of ADT, hence improving the overall quality of life for patients [18]. There was concern regarding the comparative effectiveness of IAS; however, meta-analyses show no statistical difference in time to remission between IAS and CAS [19].

Androgen also triggers the secretion of prostate-specific antigen (PSA), a protein normally found in seminal fluid. Usually, PSA is contained within the cytoplasm of prostatic acinar cells and the ductal epithelium. However, when the prostate becomes cancerous, PSA can leak into the bloodstream via disruption in the epithelial wall. PSA is detectable in the serum, and an elevated level of PSA is a strong indicator of prostate cancer presence and growth. Hence, clinicians can monitor prostate cancer progression using longitudinal measurements of PSA. However, the correlation between PSA levels and tumor volume is imperfect and can vary over time due to phenotypical and physiological changes in the cancer itself [20]. On the other hand, PSA measurements can be taken frequently and at a low cost, so they remain an indispensable tool that can be used to gain valuable insights into the dynamics of the tumor [17,21].

The development of dynamical models for prostate cancer dates back almost two decades [22]. Since then, there has been an array of models developed to study different aspects of prostate cancer and its treatments [12,13,16,23,24,25,26,27,28,29,30,31,32,33,34,35,36,37,38,39,40,41]. Most of these results have been reviewed and synthesized previously [5,42,43].

In this study, we use a mathematical model to track the clonal evolution of prostate cancer cells. Using longitudinal measurements of androgen and PSA from a clinical trial for IAS, we demonstrate the potential of two new methods for predicting an imminent treatment failure due to the growing dominance of resistant cellular strains [17]. The first method depends on the mathematical model, while the second is model-free yet still based on the underlying theoretical implication of the first method. In this work, we classify all prostate cancer cell types into two broad categories: those susceptible and those resistant to ADT. For these two indicators of treatment resistance, or biomarkers, each has its own advantages, but both have the potential to be useful in clinical settings.

## 2. Materials and Methods

Our primary investigative tool is a mathematical model based on the Droop cell quota framework and multi-species competition theory [44,45,46]. We present the model in detail in the quick-guide box. Several important iterations of this model are included in the Appendix A. In summary, the model represents a system wherein androgen is produced and secreted into the blood, before diffusing into the intracellular spaces of cancerous cells in the prostate. The resulting cell quota of androgen, *Q*(*t*), is representative of the bound androgen receptors, which drive the proliferation and apoptosis of prostate cancer cells. In order to proliferate, cancer cells require a certain amount of bound androgen receptors. This minimum number of bound receptors is called the minimum cell quota, or q, within the Droop framework. For example, if a cell lacks sufficient bound receptors to support proliferation (Q(t)≤q), then the proliferation term becomes zero. The Droop functions have been used extensively to model prostate cancer [16,23,24,25,26,27].

As prostate cancer cells proliferate, they produce PSA. Previous studies considered the PSA production rate to be linearly dependent on the current amount of bound androgen receptors. To be more biologically realistic, we assume that the PSA production rate is proportional to the proliferation rate of cancer cells. That is, we assume if cancer cells do not proliferate, then they do not produce PSA. We test these two assumptions to show that our proposed alternative PSA production rate better captures the qualitative behaviors of PSA dynamics (see Appendix A). These considerations for our modeling framework are highlighted in Figure 1.

### Model Quick-Guide Box

This model is built on previous work by Kuang et al. [16,23,24,25,26,27]. A schematic is provided in Figure 1.

The total volumes of cancer cells susceptible to treatment (Castration Susceptible or CS) and resistant to treatment (Castration Resistant or CR) are represented by $x_{1}$ and $x_{2}$, respectively. Hence, together $\frac{dx_{1}}{dt}$ and $\frac{dx_{2}}{dt}$ capture the rate of change of the total cancer population as it undergoes intermittent androgen suppression therapy (IAS). Our model of this dynamic takes the following form:(1)$$\frac{dx_{1}}{dt}=\underset{proliferation}{\underbrace{max\left\{\mu \left(1-\frac{q_{1}}{Q}\right)x_{2,}0\right\}}}-\underset{death}{\underbrace{dx_{1}\left(x_{1}+x_{2}\right)}}-\underset{transformation}{\underbrace{c\left(\frac{K}{Q\;+\;K}\right)\;x_{1,}}}\;$$
(2)$$\frac{dx_{2}}{dt}=\underset{proliferation}{\underbrace{max\left\{\mu \left(1-\frac{q_{2}}{Q}\right)x_{2,}0\right\}}}-\underset{death}{\underbrace{dx_{2}\left(x_{1}+x_{2}\right)}}+\underset{transformation}{\underbrace{c\left(\frac{K}{Q+K}\right)x_{1}}},$$

The maximum functions represent androgen-dependent proliferation. In the presence of sufficient androgen (i.e., $Q>q_{i}$), proliferation and PSA production rates are positive. Otherwise, when androgen is below the minimum cell quota level, proliferation and PSA secretion rates fall to zero. Additionally, androgen also affects the transformation rate based on a standard saturation.

The density-dependent death term represents the competition between and among the $x_{1}$ and $x_{2}$ populations for other resources, including glucose and oxygen. We may select different competition rates d to reflect the cost of gaining resistance and account for evolutionary trade-offs [13,28,29]. The current model assumes, however, that even without treatment, the tumor is guaranteed to become treatment-resistant given sufficient time. This is because, for the sake of simplicity, we purposefully neglect the cost of resistance and the variety of potential subclones. This omission is justified because the modern theory of treatment, that is, “hit hard, hit fast”, results in the total destruction of all susceptible clones [42]. This model formulation is suited to our goal of investigating treatment resistance under current clinical practice. A model with an explicit competition rate would be required to explore adaptive therapy [13,29,41].

A crucial component of our modeling framework is the pair of minimum cell quota parameters $q_{1}$ and $q_{2}$ that define the threshold amounts of bound androgen receptors required for the proliferation of the two subclones. While we elect to model only two subpopulations, the classification is based on a population average, so the resistance level of $x_{2}$ may change over time to account for evolutionary factors. In particular, we expect the resistant population $x_{2}$ to demonstrate a decreasing average dependence on androgen as the treatment continues. This means we expect to see a diminishing *q*_2_ as the model calibrates its value over the course of treatment (Figure 1b). This means $x_{2}$ represents the currently dominant resistant clone.

Under selective pressure of the treatment, cells may adapt to increase their survivability (i.e., mutations, genomic or epigenetic changes). The transformation term represents the rate at which cells adapt and become more independent of androgen over time. Previous modeling studies have shown that it is not necessary to include a transformation term from resistant to sensitive phenotypes [25]. The parameter *K* determines how sensitive this transformation rate is to the level of bound androgen receptors.

The free androgen and the bound androgen receptors are represented by A(t) and Q(t), respectively. We model their dynamics as follows:(3)$$\frac{dA}{dt}=\underset{primary\;production}{\underbrace{\gamma_{1}u\left(t\right)\left(1-\frac{A}{A_{0}}\right)}}+\underset{secondary\;production}{\underbrace{\gamma_{2}}}-\underset{degradation}{\underbrace{\delta A}}\;$$
(4)$$\frac{dQ}{dt}=\underset{diffusion}{\underbrace{m\left(A-Q\right)}}-\underset{used\;up\;by\;cancer\;cells}{\underbrace{\frac{\mu \left(Q-q_{1}\right)x_{1}+\mu \left(Q-q_{2}\right)x_{2}}{x_{1}+x_{2}}}}\;$$

Production of free androgen follows a negative feedback loop where the maximum rate parameter is $\gamma_{1}$ and the homeostasis androgen level is $A_{0}$. Testicular production of androgen (primary production) is intermittently suppressed by the administration of ADT, which is represented by the Heaviside function u(t). The rate of adrenal androgen production $\gamma_{2}$ is constant and fixed at a small percentage of the primary production$\;\gamma_{1}$. Serum androgen concentration degrades at constant-per-capita rate δ.

Free androgen diffuses into cells and binds to androgen receptors at a maximum rate m. We assume that androgen receptor binding happens instantaneously when androgen enters the cell. The final term in $\frac{dQ}{dt}$ is motivated by the laws of conservation, and accounts for intracellular androgen that is consumed by the cancer to fuel its proliferation [25].

In general, existing prostate cancer models are built on the assumption that the rate of PSA production is a linear function of the amount of tumor cells [5]. However, PSA production is intrinsically linked to cancer proliferation via the same transcription factor: bound androgen receptors (Figure 1a). We incorporate this observation into our model by formulating the PSA production rate as a function of cellular activity. In addition, we add a baseline production of PSA due to healthy prostate cells. These assumptions lead to the following equation for PSA dynamics:(5)$$\frac{dP}{dt}=\underset{baseline}{\underbrace{bQ}}+\underset{PSA\;production\;by\;x_{1}}{\underbrace{max\left\{\sigma_{1}\left(1-\frac{q_{1}}{Q}\right)x_{1,}0\right\}}}+\underset{PSA\;production\;by\;x_{2}}{\underbrace{max\left\{\sigma_{2}\left(1-\frac{q_{2}}{Q}\right)x_{2,}0\right\}}}-\underset{PSA\;pclearance}{\underbrace{\epsilon P}}\;$$

One desirable consequence of connecting the cellular proliferation function to the PSA production rate is the explicit connection between minimum cell quotas, $q_{1}$ and$\;q_{2}$, and the level of PSA. As the neoplasm becomes increasingly indifferent to environmental androgen, resistant cancer cells should secrete PSA more freely even during active ADT. We hypothesize that in our model the dynamics of PSA will be sensitive to changes in the *q*_1_ and *q*_2_ parameters. Additionally, the model should reflect the divergence between PSA and androgen levels observed in later cycles of resistant patients.

Table 1 contains a summary of model parameters and ranges, and additional information on the model formulation and its parameters can be found in the Appendix A.

We fit our model to patient data from a clinical study of IAS at the Vancouver Prostate Centre [17]. The data contain longitudinal measurements of PSA and androgen for 71 patients during IAS. Also recorded is information on the ultimate result of each patient’s treatment.

We use MATLAB 2021a to perform our simulation and analysis. In particular, we use MATLAB function *fmincon* to fit the model to patient data. To limit potential issues of parameter identifiability, we only estimate four key parameters [27]. To do so, we fit the model against segment data, one for every on or off-treatment period. The remaining parameters are fixed to values determined by a test run performed over the first two cycles of data. Additionally, we apply more weight to the discrepancy between the model simulation and PSA data as compared to androgen data (80% to 20%, respectively). Weighted error approaches have been shown to improve overall model fitting [26]. The Appendix A contains additional details regarding the data used, the method of calculating error, and other considerations of the model fitting.

We present two potential biomarkers that may be used to predict the development of resistance to ADT. Our first predictive proposed indicator is the ratio between initial and final (most recent) values of$\;q_{2}$. Selective pressure during each treatment cycle causes the resistant subclones to become less dependent on environmental androgen through a variety of mechanisms [47]. Therefore, we expect the value of $q_{2}$ to decrease over sequential estimates. We aim to determine, using clinical data fitted to the model, if one can define a threshold that is correlated with an increased probability of treatment failure.

Treatment resistance can also be recognized in the data by the divergence between androgen and PSA dynamics. Initially, when a patient begins androgen suppression therapy, the PSA and androgen behavior are qualitatively similar, rising and falling together as treatment is applied and removed. However, as the neoplasm becomes more castration-resistant, more cells survive the cycles of androgen deprivation to secrete PSA regardless of treatment status. The second proposed biomarker follows from these observations by taking the ratio of serum measurements of androgen and PSA to be an indicator of treatment failure. This second biomarker is related to the first in that the parameter $\;q_{2}$ is directly correlated with resistance. A low estimated $\;q_{2}$ therefore represents a clonal population’s ability to proliferate even when deprived of androgen, which would be reflected by relatively high and low measurements of PSA and androgen, respectively. In essence, the second proposed biomarker is a model-free form of the first biomarker.

In order to evaluate the predictive potential of both proposed biomarkers, we first sorted patients into two groups: success or failure. We defined failures as discontinuations due to resistance or death from prostate cancer; we classified all other outcomes as successes. Next, we sought to identify a correlation between the biomarker values and treatment success or failure. Distinct thresholds were established for each ratio that, when reached, signify treatment failure in the next cycle.

We present here two sets of thresholds calculated with two different methods. The first method determines thresholds that maximize the accuracy of our predictions based on the Vancouver dataset. These are denoted as the Max thresholds. The second method uses MATLAB’s support vector machine function *fitcsvm* to generate thresholds that, while less accurate than the Max threshold when used with the current dataset, are potentially more accurate across a larger cohort of patients. The thresholds calculated by the second method are referred to as the SVM thresholds. We then test the robustness of these thresholds with cross-validation.

## 3. Results

In general, the model closely captures PSA and androgen dynamics (Figure 2). However, its fit favors PSA data over androgen data [26]. The model fit is sufficiently reasonable to generate accurate predictive biomarkers. In Figure 2, we demonstrate the model’s ability to fit data and the subsequent implications regarding changes in the castration-sensitive and resistant cancer subpopulations (right column figures). After the initial treatment, the castration-sensitive cancer subpopulation is essentially replaced by the castrate-resistant cancer subpopulation. However, because we use a dynamic value for the level of resistance, the emergence of the castration-resistant cancer subpopulation does not signify treatment failure. For instance, in Figure 2b, the castration-resistant population still declines significantly during the second treatment period, which corresponds to the second phase of androgen decrease (occurs between day 350 and day 700). This implies that the castration-resistant subpopulation’s level of resistance is still developing, and not yet sufficient for it to be unaffected by androgen suppression. On the other hand, during the third treated cycle (starting around day 800], the castration-resistant cancer subpopulation develops sufficient resistance to the androgen suppression therapy for it to proliferate even in the absence of androgen. In Figure 2a the castration-resistant population is mostly unaffected by the second treatment period (occurs between day 350 and the end of treatment). However, in this case, the cancer never develops sufficient resistance for the CR subpopulation to proliferate in the absence of androgen.

Figure 3 summarizes the analytical result for the $q_{2}$ ratio as a potential predictive biomarker. The result demonstrates that when the value of the $q_{2}\;$ratio exceeds either the SVM or Max threshold, it strongly indicates an impending treatment failure due to the development of resistance. The $q_{2}$ Max and SVM thresholds classify the data with accuracies of 87.3% (sensitivity: 96.1%, specificity: 65%) and 81.7% (sensitivity: 98.0%, specificity: 40%), respectively.

Figure 4 shows a summary of the analytical result for the androgen to PSA ratio as a potential predictive biomarker. The initial values of the androgen/PSA ratio are highly variable across all patients. In the early stages of treatment, there is no correlation between the ratio and that treatment’s ultimate outcome, which is consistent with the selection criteria of the patients for the clinical trial [24]. However, that is not the case if the androgen to PSA ratio is calculated using the mean values of the patient’s final on-treatment cycle. Having done this, we see that when the androgen to PSA ratio falls below the Max or SVM thresholds, it strongly indicates impending treatment failure due to the development of castration resistance. For the androgen to PSA ratio, the Max threshold is 0.19 and classifies patients with 88.7% accuracy (sensitivity: 90.2%, specificity: 85.0%), while the SVM threshold of 0.30 classifies patients with 84.5% accuracy (sensitivity: 82.4%, specificity: 90.0%). It is worth emphasizing that the androgen/PSA biomarker is calculated using only serum androgen and PSA data, and therefore does not require a mathematical model to estimate.

To test the robustness of all thresholds, we used five-fold cross-validation. For the SVM thresholds, we used the built-in MATLAB cross-validation function. The predictive accuracy for the *q*_2_ ratio SVM threshold was 79%. The accuracy for the androgen/PSA SVM threshold was 85%. For the Max thresholds, we randomized the data, performed the five-fold cross-validation, and then replicated the procedure 100 times. The mean accuracies of the *q*_2_ ratio SVM thresholds were 87% for the training sample and 84% for the holdout sample. The mean accuracies for the androgen/PSA SVM thresholds were 89% for the training sample and 85% for the holdout sample.

## 4. Discussion

Perhaps the most troublesome cancer characteristic, when it comes to treatment, is its ability to quickly adapt and evade initially effective therapies. Due to genomic instability, new cellular variations are continuously appearing, competing, and going extinct, which leads to increasing malignancy via natural selection [6,7,10]. Therefore, in many cases, it is only a matter of time before tumors evolve resistance to conventional treatments. Treatments fail when susceptible subclones have perished and replaced by resistant competitors, thus removing any possibility of continued success, even as a second-line measure. If clinicians could detect incipient resistance in advance, it would allow them an opportunity to change tactics that may lead to better clinical outcomes. Such an ability would likely improve long-term management and individualized treatment plans for cancer patients [13,28,29].

In this study, we propose two different tools that can be used to accurately determine approaching castration-resistance during intermittent ADT. Other prediction tools do exist that assess clinical prostate cancer, including AUC measures of PSA to diagnose clinically significant disease [48] and the Gleason score, which has some power to measure prostate cancer aggressiveness and predict treatment outcome [49]. However, neither PSA nor Gleason score are typically used to make real-time predictions of cancer response to treatment. In contrast, our two proposed indicators both have high predictive accuracies of 87–89%, and they rely on measurements of PSA and androgen, which are both relatively simple to obtain in real time. This observation supports their potential usefulness in the clinical setting and warrants further investigation.

Both proposed biomarkers are developed from the same classical ecological theories. However, they are calculated differently. While the androgen to PSA ratio was motivated by a mathematical model, the model is not required for its implementation; it relies entirely on serum data that can be collected as part of the standard monitoring routine. However, the model-free biomarker requires consistent, regular measurements of both androgen and PSA serum for calibration and prediction accuracy. In contrast, the $q_{2}$ ratio may be estimated using only longitudinal PSA data and baseline serum androgen level with the aid of the mathematical model. Furthermore, since $q_{2}$ reflects the degree of resistance and may be estimated independently of androgen data, it is possible to detect approaching treatment failure during the off-treatment period prior to resuming treatment. Both proposed biomarkers can be used in conjunction to improve the overall accuracy.

In our analysis, we calculate the ratios using data from the last on-treatment cycle to show the predictive potential of our proposed biomarkers. This leaves an important question unanswered: how early can these indicators predict treatment failure with sufficiently high accuracy? We demonstrate that the proposed biomarkers do not indicate treatment failure at the start of treatment and predict treatment failure with high accuracy only at the last cycle. Furthermore, we show that there is an increasing trend in $q_{2}$ ratio over each treatment interval (see Appendix A). Furthermore, we show that this ratio tends to increase with each successive treatment interval (Appendix A). On a theoretical level, this implies that as a patient is treated with IAS their ratio will increase until it eventually surpasses the thresholds that signify treatment failure. Thus, we can potentially predict the treatment’s outcome during its administration. Subsequent studies are needed to evaluate this possibility. 

We provide the summary statistics of PSA and androgen level for both groups in the Appendix A, which shows that resistant patients have a lower level of androgen at the end of treatment compared to the respondent group. This difference is the reason for the distinct androgen/PSA ratios between the two groups. We speculate that this is because the drugs used in this particular clinical trial affect each patient differently. In particular, leuprolide acetate can overstimulate the pituitary gland to produce gonadotropic cells carrying luteinizing hormones, which increases the production of androgen in the testes; however, over time, leuprolide desensitizes the pituitary gland, which ultimately ceases androgen production in the testes. This desensitization effect remains for some time after the treatment stops [16]. Thus, it is possible that, in the resistant patients, after each treatment cycle, the desensitization effect lasts for much longer leading to a low level of androgen for an extended duration. The longer duration of low androgen level prolongs the selective pressure, which can lead to an increased selection for the resistant cancer population in these patients [50].

## 5. Conclusions

The accuracy of these two biomarkers in our analysis supports the growing trend of implementing mathematical models in clinical studies [5,42]. Furthermore, our analysis reemphasizes the importance of careful data collection during treatment. The dataset that we use here contains consistent longitudinal measurements of PSA and serum androgen for each patient over several years of treatment. However, this quantity of data is not often collected in practice. For these or any, biomarkers to have practical value, blood panels measuring serum PSA and androgen must be taken regularly and consistently. This would maximize the usefulness of any associated mathematical models [27].

We remark that even while very high/low levels of PSA and testosterone can affect their ratio, this qualitative observation is still consistent with our result. If the PSA level is high when the androgen level is high, their ratio may be higher than the threshold that determines treatment failure. On the other hand, if the PSA level is very high while the androgen level is not, then the ratio is naturally going to be smaller, which is more indicative of treatment failure. Finally, while PSA and androgen (but mostly PSA) are used as an indication of treatment failure, there would still be a requirement for confirmation by other means such as a radiographic scan. Our study shows the potential of using both PSA and androgen quantitatively for personalized treatment.

In summary, the development of mathematical models in clinical settings can benefit tremendously from incorporating datasets that are specifically designed and collected for the validation of those models.

## Figures and Tables

**Figure 1 cancers-14-04033-f001:**
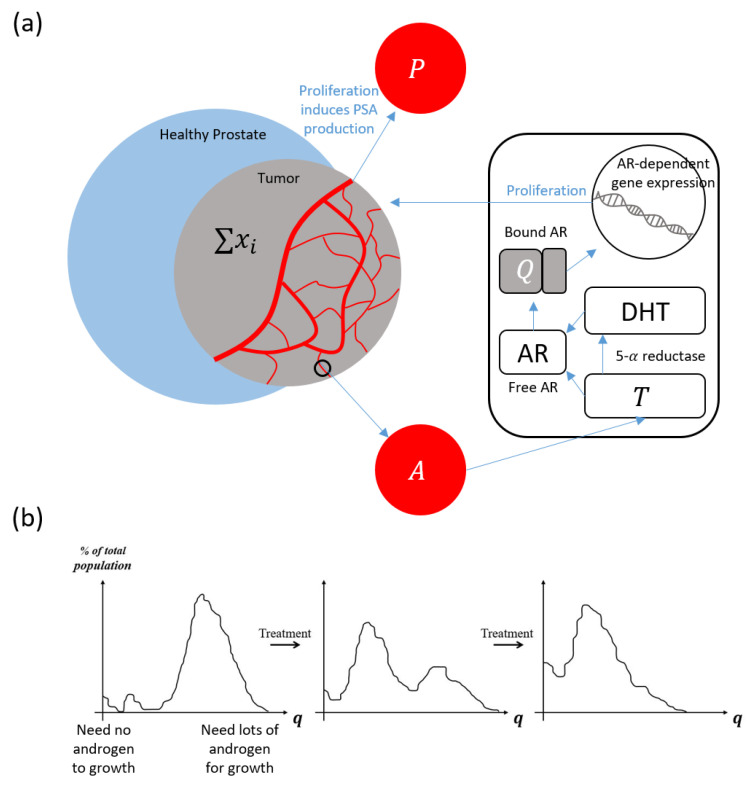
Schematics of model foundation and evolutionary framework. (**a**) Androgen (testosterone) enters the cancer cells. Some is converted to the potent dihydrotestosterone (DHT) with the help of 5-α reductase. Both then bind to the androgen receptors (AR). The bound androgen receptors send proliferative signals for cancer to grow and produce PSA (P). PSA then leaks into the bloodstream. (**b**) The distribution of the minimum cell quota (q) prior to treatment, q profile skews toward higher values of q. This means most cancer cells are initially sensitive to treatment. During each treatment, this evolutionary landscape shifts toward a lower average q, meaning an increasing number of cells become less dependent on exogenous androgen.

**Figure 2 cancers-14-04033-f002:**
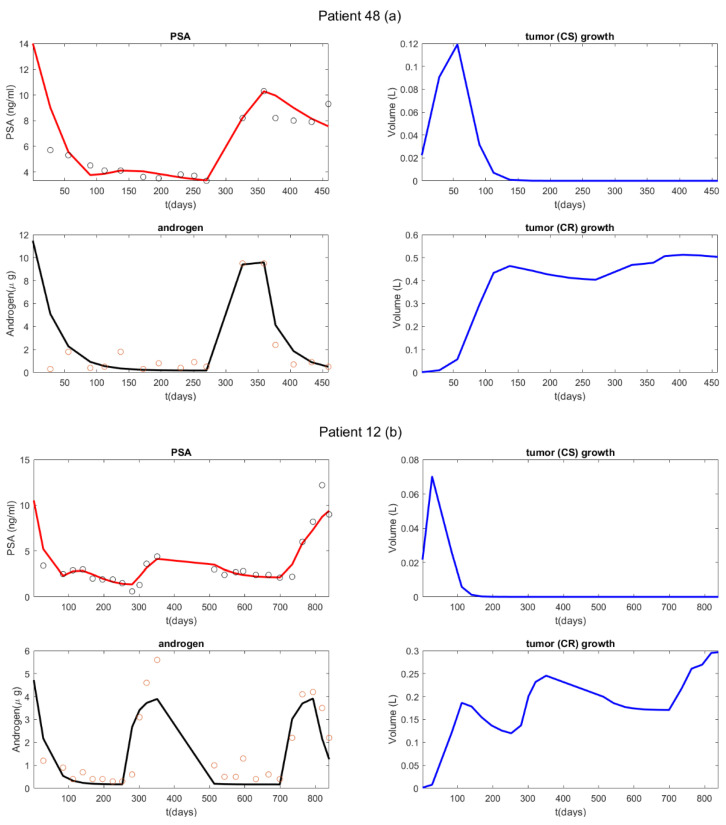
Model validation: Best-fit model solutions to the dynamics of serum androgen and PSA levels. Circles represent patient measurements, and the solid lines are solutions of model (model equation number). ‘CS’ = castration susceptible tumor cell population; ‘CR’ = castration-resistant population. Panel (**a**) was produced by a short dataset 1.5 cycles long, and panel (**b**) by a dataset 2.5 cycles long.

**Figure 3 cancers-14-04033-f003:**
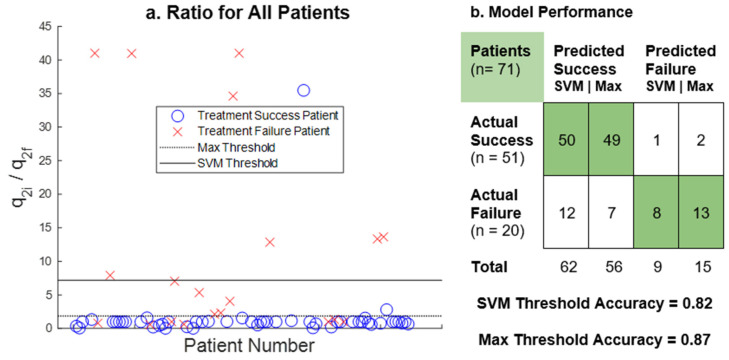
The predictive potential of the q_2_ ratio: The scatterplot (**a**) indicates the value of the *q_2_* ratio for every patient in the dataset. The ratio is between the initial and final values of the *q_2_* parameter calculated by the mathematical model. Max (dotted line) and SVM (solid line) threshold values are shown. The confusion matrix (**b**) compares actual patient outcomes with outcomes predicted by *q_2_* ratio with respect to the thresholds.

**Figure 4 cancers-14-04033-f004:**
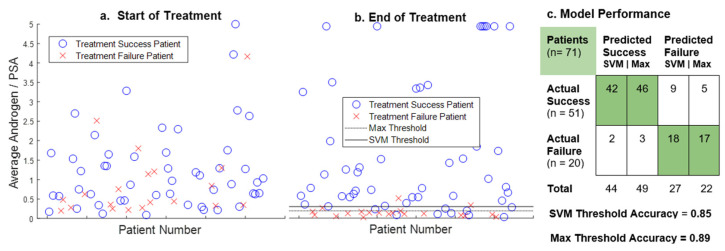
The predictive potential of the Androgen/PSA ratio: Scatterplot (**a***)* shows the value of the androgen/PSA ratio for every patient when calculated using mean values of androgen and PSA from the first 200 days of treatment. Scatterplot (**a**) demonstrates that there is little correlation between the value of the ratio and treatment outcome when calculated in this manner. Scatterplot (**b**) shows the same ratio calculated using mean androgen and PSA values from the patient’s final on-treatment cycle, not exceeding 200 days. For the purposes of this figure, all ratio values greater than five are represented as five. Scatterplot (**b**) shows two thresholds below which values of the androgen/PSA ratio indicate impending treatment failure. The confusion matrix (**c**) compares actual patient outcomes to outcomes predicted by the ratio with respect to the two thresholds.

**Table 1 cancers-14-04033-t001:** Parameter definitions and boundaries: This table describes the physiological interpretations of the fifteen parameters used in this model [5,26]. The range column indicates the upper and lower bounds within which an error-minimizing function may establish an optimal value with respect to a concrete set of patient data. The * in place of upper and lower bounds of $A_{0}$ is because the range of $A_{0}$ is patient specific and is set to the patient’s maximum recorded androgen data ±10.

Parameter	Description	Range	Unit
Μ	max proliferation Rate	0.001–0.09	[day]^−1^
*q_1_*	$\mathrm{minimum}\;\mathrm{cell}\;\mathrm{quota}\;\mathrm{for}\;x_{1}$ to proliferate	0.41–1.73	[nmol][day]^−1^
*q_2_*	$\mathrm{minimum}\;\mathrm{cell}\;\mathrm{quota}\;\mathrm{for}\;x_{2}$ to proliferate	0.01–0.41	[nmole][day]^−1^
*d*	density death rate	0.001–0.30	[L]^−1^[day]^−1^
*c*	maximum mutation rate	0.00015–0.00015	[day]^−1^
*K*	half-saturation constant for mutation	1–1	[nmole][day]^−1^
$\gamma_{1}$	androgen production by testes	0.008–0.8	[nmol][day]^−1^
$\gamma_{2}$	androgen production rate by adrenal gland	0.005–0.005	[nmol][day]^−1^
$A_{0}$	homeostasis serum androgen level	*	[nmol]
Δ	androgen degradation rate	0.03–0.15	[day]^−1^
*b*	baseline PSA production rate	0.0001–0.1	[μg][nmol]^−1^[day]^−1^
$\sigma_{1}$	$\mathrm{maximum}\;\mathrm{PSA}\;\mathrm{production}\;\mathrm{rate}\;\mathrm{by}\;x_{1}$	0.001–1	[μg][nmol]^−1^[L]^−1^[day]^−1^
$\sigma_{2}$	maximum PSA production rate by *x_2_*	0.001–1	[μg][nmol]^−1^[L]^−1^[day]^−1^
ϵ	PSA clearance rate	0.0001–0.1	[day]^−1^

## Data Availability

All codes and results are available online: https://github.com/allison-weber/prostate_cancer_modeling_study (accessed on 7 January 2022).

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
