# Peer review of "High Accuracy Indicators of Androgen Suppression Therapy Failure for Prostate Cancer—A Modeling Study"

_cancers, 2022, doi:10.3390/cancers14164033_

Round 1
Reviewer 1 Report
I can strongly recommend this paper for publication. It is well written and the research is highly translatable.
I have only a few minor comments:
1. Page 2, Line 67 - do you mean intercellular competition instead of intracellular?
2. Page 3, Lines 129-131 - if you claim that non-proliferating cells do not express PSA as biologically realistic, then this must be supported with a citation from the biology side. Otherwise it should be a model assumption.
3. Page 6, Line 211 - Are these patients in fact put on IAS with intact prostates? If not, why should there be a baseline PSA production? In general, ADT is usually started only after a radical prostatectomy.
4. One question about using change in q2 as predictive of treatment failure is how early can failure be predicted? That is, can you predict failure in the previous cycle before the patient actually fails? Perhaps this is already done but is not so clear what is meant by 'impending' failure, in quantitative terms.
Reviewer 2 Report
In this research paper, Meade and colleagues introduce two methods to improve the accuracy of classical biomarkers in identifying prostate cancer patients who are unresponsive to hormone therapy.
Hormone deprivation therapy is the standard treatment for advanced / metastatic prostate cancer or extraprostatic disease.
However, it is difficult to determine initial reactivity or the acquisition of resistance during the course of treatment.
For this reason, the identification of biomarkers capable of distinguishing the responder from the unresponsive patients is a major challenge in the field of prostate cancer management.
Here the authors proposed two predictive models using the patient's PSA and androgen data from a clinical trial of intermittent treatment with androgen deprivation therapy.
The first model is calculated using a mathematical model that predict the presence of resistant cell clones and the second is based on the ratio of blood testosterone and PSA levels.
The results of this study suggest that the model proposed by the authors is able to discriminate at the end of treatment who has responded well to hormonal therapy and who has not.
The authors would like their model to be used to monitor patients on treatment to identify early when resistance develops. However, this is the authors' wish and not evidence. The text of the paper must be clear about what has been demonstrated and what a future assumption or perspective is.
In Figure 2, no panels a and b are displayed as indicated in the caption.
The panels on the right showing a tumor susceptible to castration or resistant to castration are very difficult to understand, as there are no references to this figure in the text. What do they represent? Please describe better the meaning of Figure 2.
Why do the authors use the androgen / PSA ratio as an indicator of therapy success / failure?
Metastatic prostate cancer is commonly characterized by very high levels of PSA in the blood. If the PSA value is the denominator of the ratio it is obvious that a very high PSA causes a very low ratio and vice versa.
Likewise the testosterone value range is highly subjective and very high / low levels could profoundly affect the ratio value.
Commonly, the decrease in both PSA and androgens is considered a success parameter, but in many cases a decrease in PSA is sufficient to determine its reactivity.
Can the authors show PSA and testosterone values ​​(mean, range, median IQR) for the respondent and resistant group before and after treatment?
The authors argue that the two models could be combined to give a greater predictive value. This is not obvious. To support it, it is necessary to evaluate whether the parameters are directly related or not. Please rearrange this conclusion.
Minor point:
In Figure 4 panel A there is an error in the legend.
Round 2
Reviewer 2 Report
I appreciate the efforts of the authors in reviewing the article.
However, several points remain unclear.
1. As noted in the first review, the authors cannot assume that the two parameters considered in their work can be combined to show greater predictive capacity, simply by saying "The high sensitivity of the first indicator and the high specificity of the second indicator mean 38 they can complement one another in clinical settings.".
To state this, the authors should at least evaluate whether the two parameters have significantly different specificities at fixed sensitivity. If so, this evidence suggests that they can potentially complement each other. However, I'm pretty sure that for androgen / PSA if we increase the sensitivity we get a lower specificity.
In case the authors wish to keep the abstract sentence as it is, they can demonstrate this by a simple regression/correlation analysis.
Please provide proof of their assumption or change the sentence.
2. I thank the authors for adding a new graph in the supplementary materials (Fig 9s) to display PSA and testosterone values ​​(mean, range, median IQR) for the respondent and resistant groups before and after treatment.
How do they explain/interpret the lower testosterone levels in the resistant group compared to the respondent?
Round 3
Reviewer 2 Report
I appreciate the current revision.